# Effects of nurses' shiftwork characteristics and aspects of private life on work-life conflict

**Hye-Kyung Oh**[1]*, **Sung-Hyun Cho**[2]

1 Department of Nursing, Division of Nursing and Public Health, Daegu University, Daegu, South Korea,
2 College of Nursing, Seoul National University, Seoul, South Korea

* katie5@daegu.ac.kr

## Abstract

### Background

As nurses work highly irregular hours, the characteristics of shiftwork and aspects of their private lives are important factors that may contribute significantly to work-life conflict.

### Purpose

This study examined the effects of nurses' shiftwork characteristics and aspects of their private lives on work-life conflict.

### Methods

The participants included 271 registered nurses working three-shift rotations in five types of units at four hospitals in South Korea. We distributed structured questionnaires regarding shiftwork characteristics, private life, and work-life conflict. Data were analyzed using multiple linear regression analysis.

### Results

The significant factors relating to work-life conflict included control over shift start and finish times ($\beta = -0.16$, $p = .019$), frequency of swapping shifts with colleagues ($\beta = 0.15$, $p = .025$) among shiftwork characteristics, and leisure constraints ($\beta = 0.39$, $p = < .001$) in aspects of private life.

### Conclusion

Plan and policies for improving nursing environments should focus on improving nurses' control over shiftwork and decreasing leisure constraints.

**Data Availability Statement:** All relevant data are available from protocols.io: https://www.protocols.io/view/effects-of-nurses-shiftwork-characteristics-bhihj4b6/metadata.

**Funding:** The authors received no specific funding for this work.

**Competing interests:** No authors have competing interests.

# Introduction

## Background

Nurses are a frontline occupational group who work with patients throughout the daily 24-hour period. In Korea, hospital nurses work three shifts [1]. The shifts are classified into days, afternoons, and nights. Working irregular hours frequently, however, could severely impact work-life balance [2, 3]. Adjusting work schedules can be a solution to improve work-life balance [4]. Currently, there exists a serious shortage of nurses in small and medium-sized hospitals in South Korea [5]. This does not mean that there is necessarily an overall shortage in the total number of nurses. However, there is mismatch in the supply and demand of nurses. To resolve this situation, the Korean Ministry of Education proposed an alternative plan: to transfer university students to nursing colleges to increase the number of nursing students from 10% to 30%. Under this scheme, students who wanted to pursue nursing courses at a nursing college were transferred [6]. However, the critical problem is that a large number of registered nurses are currently out of employment. Therefore, the fundamental causes leading to the mismatch in supply and demand of nurses have to be identified and rectified.

The primary reason for the large proportion of idle nurses is irregular working hours such as three-shift schedules. It is unsuitable for pregnancy nurses and nurses with children who have to provide childcare. This is also impacted by social environments where maintaining work-life balance becomes difficult [7]. Shiftwork is inevitable to ensure continuous, high-quality nursing care. However, there are limitations to adjusting shiftwork schedules of the limited numbers of nurses in a way that reflects a variety of personal preferences. As optimal shiftwork schedules that satisfy everyone have not been developed, it is important to obtain basic data to identify the facts associated with shiftwork characteristics and to identify the relevant variables that should be incorporated into policy suggestions that can improve the current situation.

In the USA, the National Institute for Occupational Safety and Health had conducted three waves of studies from 1993 to 2006 on various issues related to nurses, including shiftwork characteristics [8]. As part of the studies, shiftwork characteristics were explored using items of the Standard Shiftwork Index (SSI) in the form of a questionnaire used internationally. Using the data from these studies, researchers looked into conditions of working hours, overtime, etc. and also analyzed non-work activities such as housework, childcare chores, and leisure time [8]. As such a study has not been conducted in Korea, we deemed it necessary to analyze the shiftwork characteristics and aspects of the private life of nurses in the country.

In previous studies, nurses' working hours, overtime [9, 10], and flexible working hours [9] were found to be related to work-life conflict. Irregular shiftwork schedules and long working hours were found to be the most important factors in increasing work-life conflict [10].

Childcare and time spent on domestic chores were activities of private life that were found to interfere with work. While there have been many studies in various countries incorporating objective variables such as shiftwork schedule analysis and work pattern investigations, most of the studies in Korea have been based on nurse recognition surveys. Hence, the general aim was to identify the factors affecting work-life conflict of nurses. Taking these factors into account, we recommend that plans and policies should be implemented to improve the environment of the three-shift schedules of the nurses.

# Materials and methods

## Setting and sample

This study was approved by the bioethics committee of Seoul National University (IRB NO. 1909/001-001). Convenience samples of 271 shift nurses were selected from four secondary

care hospitals with medical, surgical, intensive care, emergency, and integrated nursing care units. These secondary hospitals (comprising over 300 beds) located in close proximity areas were selected to minimize variation among hospitals. The participants were recruited per ward unit, with the cooperation of the nursing departments and hospitals. The participants were selected based on their willingness to participate in the study voluntarily. The inclusionary criteria involved (a) shiftwork nurses, (b) who were working at the designated hospitals. Nurses were excluded if they were newly nurses recruited under training, and could not work independently. The sample size was calculated using G*power 3.0.10 based on a significance level of 0.05, a medium effect size (0.15), a power of 0.95, and 11 independent variables. According to this analysis, the minimum sample size was 178, although 280 were ultimately recruited with consideration of the dropout rate. After the completed questionnaires were received, a preprocessing check was conducted to remove incomplete or invalid data.

**Variables and instruments.** Shiftwork characteristics were measured using 11 variables. Some of these variables were measured using the SSI developed by Barton. et al. [11]. Other variables that were measured using items relating to shiftwork characteristics are provided in Table 1. Since the SSI is based on self-reporting, concerns have been raised about inaccuracies and response bias relating to participants' memory. To compensate for this, in the current study, unit shiftwork schedules were directly verified by the researcher. Variables that the researcher directly collected data on the included the number of 40-hour plus working weeks in a month, unsocial working hours per week, number of maximum consecutive shifts worked during the previous three months, number of unhealthy shift patterns in a month, and nurse staffing adequacy.

**Table 1. Research variables for shiftwork characteristics, aspects of private life, and work-life conflict.**

| Categories | Variables | Explanation | Instrument |
|---|---|---|---|
| Work hours | Average number of over 40-hour working weeks in a month | Average number of nurses who are scheduled to work for over 40 hours per week | SSI |
| | Unsocial working hours per week | Unsocial work hours per week (night work: 10 pm to 5am, evening work: 6 pm to 10 pm, and Saturday /Sunday work) | European unsocial hours |
| | Overtime | Amount of overtime work per day | SSI |
| Shift patterns | Number of maximum consecutive shifts | Number of maximum consecutive shifts between days off over 3 months | SSI |
| | Average number of unhealthy shift patterns in a month | Average number of nurses who are assigned unhealthy shift patterns per month | |
| Nurse staffing adequacy | Nurse staffing adequacy (%) | Ratio of the number of nurses required in the ward to the number of nurses who actually work there. | |
| Shift stability | Control over specific shifts | 5-point scale | SSI |
| | Control of specific start & finish times | | |
| Shift instability | Required to change shifts at short notice | | |
| | Swapping shifts with colleagues | | |
| | Requests to work specific shifts | | |
| Characteristics of private life | Family status | Current living arrangements of the subject | Previous studies |
| | Demands at home | Dependent children, caring for someone with a chronic disease/handicap, caring for family/friend, housekeeping responsibility | |
| | Life events during past year | presence of life events | |
| | Time spent on housework or child rearing | Time spent on housework or child rearing on work /holidays | |
| | Support at home | Whether domestic help is available or not | |
| Leisure constraints | Leisure constraints | Intrapersonal/interpersonal/structural constraints | Leisure constraints |
| Work-life conflict | Work-life conflict | (Time based) Work interference with life, Life interference with work | Work-life conflict |

To verify the average number of unhealthy shift patterns in a month, different types of unhealthy working patterns were enumerated by the researcher with reference to health management for shift workers of the Korean Ministry of Employment and Labor [12], and the opinion on recommendations for working guidelines for healthy shift workers of the Korean Confederation of Trade Unions [13]. After these items were developed, three professors of nursing management and three nurses with over ten years of clinical experience conducted a content validity survey (CVI), to arrive at eight unhealthy shift patterns comprising items with validity scores of 0.8 or higher. These items were: double shifts; working 3 shift rotations for more than five consecutive days; working more than seven nights in a month; working more than three consecutive night shifts; returning to work without a 48-hour rest period after night shifts, less than 24 hours rest period between shifts; less than two nurses working on night shifts; and backward rotating working system.

The formula for calculating nurse staffing adequacy was the sum of the total number of nurses assigned during a 24-hour working day divided by the total number of required nurses for 24 hours of the same day. The number of nurses required for each unit was calculated according to the Korean graded fee of nursing management, the criteria for deploying staffing in the integrated nursing care unit, and the number of nurses in emergency rooms for each emergency medical institution. The calculation of the number of nurses required by each unit incorporated the three-shift rotation, 40-hour plus working per week, vacations, and days-off. Therefore, it was necessary to divide 1.6 from the originally required unit number per day in consideration of the shiftwork system and personal leave so that the number of nurses actually needed to operate the unit can be obtained [14, 15]. An example of a nurse staffing adequacy calculation is provided in Fig 1. As the hospital was a general hospital and had a second-grade fee for nursing management (general ward), the ratio of the number of nurses and beds should be between 1:2.5 and less than 1:3.0. The number of beds in ward was 54 beds. Therefore, a minimum of 19 nurses are required to satisfy the second-grade fee for nursing management. Among the 19 nurses, the required number of nurses per day was 11.875 (about 12) from the

| Nurses | TUR | FRI | SAT | SUN | MON | TUE | WED | TUR | FRI | SAT | SUN | MON | TUE | WED | Total workdays | Number of nurses assigned (A) | Required number of nurse (B) | Nurse staffing adequacy (A)/(B) |
|---|---|---|---|---|---|---|---|---|---|---|---|---|---|---|---|---|---|---|
| | 1 | 2 | 3 | 4 | 5 | 6 | 7 | 8 | 9 | 10 | 11 | 12 | 13 | 14 | | | | |
| Nurse-1 | D | D | D | OFF | D | D | D | D | E | OFF | OFF | D | D | D | 11 | 78 | 11*12=132 | 0.59 |
| Nurse-2 | OFF | E | E | E | N | N | OFF | D | D | D | E | E | E | OFF | 11 | 75 | 11*12=132 | 0.57 |
| Nurse-3 | N | OFF | D | D | E | E | E | OFF | E | N | N | N | OFF | E | 11 | 75 | 11*12=132 | 0.57 |
| Nurse-4 | E | N | N | N | OFF | D | D | E | OFF | D | D | E | N | N | 12 | 82 | 12*12=144 | 0.57 |
| Nurse-5 | D | D | D | OFF | E | E | N | N | N | OFF | OFF | D | D | D | 11 | 78 | 11*12=132 | 0.59 |
| Nurse-6 | D | OFF | OFF | E | OFF | D | D | OFF | D | D | D | D | OFF | D | 9 | 61 | 9*12=108 | 0.56 |
| Nurse-7 | OFF | D | E | N | N | N | OFF | D | D | E | OFF | N | N | N | 11 | 76 | 11*12=132 | 0.58 |
| Nurse-8 | E | E | OFF | D | D | OFF | N | N | N | OFF | E | E | E | E | 11 | 76 | 11*12=132 | 0.58 |
| Nurse-9 | N | N | N | OFF | D | E | E | E | OFF | N | N | OFF | D | OFF | 10 | 68 | 10*12=120 | 0.57 |
| # of day shifts' nurses | 3 | 3 | 3 | 2 | 3 | 3 | 3 | 3 | 3 | 3 | 2 | 3 | 3 | 3 | | | | |
| # of evening shifts' nurses | 2 | 2 | 2 | 2 | 2 | 2 | 2 | 2 | 2 | 1 | 2 | 3 | 2 | 2 | | | | |
| # of night shifts' nurses | 2 | 2 | 2 | 2 | 2 | 2 | 2 | 2 | 2 | 2 | 2 | 2 | 2 | 2 | | | | |
| Total assigned nurses | 7 | 7 | 7 | 6 | 7 | 7 | 7 | 7 | 7 | 6 | 6 | 8 | 7 | 7 | | | | |

**Fig 1. Example of nurse staffing adequacy.**

1.6 division and by considering the 3-shift rotation, 40-hour plus working per week, and holidays. In the case of Nurse 1, they worked 11 days. Therefore, multiplying the required number of nurses by 12 for each workday would result in a total of 132 nurses. The total number of nurses assigned on the workdays was 78; thus, only 59% of the required number of nurses were at work on the workdays.

Aspects of private life were measured using the item developed by Jansen, Kant, Kristensen, and Nijhuis [16], Takeuchi and Yamazaki [17], and Raymore, Godbey, Crawford, and Von Eye [18], which were modified by Hubbard and Mannell [19] and Hwang and Seo [20]. This measure consisted of five variables: family status; demands at home; time spent on housework and child rearing; support at home; and leisure constraints. Family status, demands at home, life events during past year, and support at home were measured dichotomously (yes or no). Time spent on housework and child-rearing (workday and holiday) was measured on a 5-point scale (0 to more than 7 hours for workdays; zero to more than 12 hours for holidays). Leisure constraints consisted of intrapersonal, interpersonal, and structural constraints, and were measured on a 5-point Likert scale (ranging from "not at all" to "very much").

The variable of work-life conflict was measured using a scale developed by Carlson, Kacmar, and Williams [21]. It consisted of two elements (work interference with life, life interference with work). The six items (time-based work and life conflict) included statements such as "My work keeps me from my family activities more than I would like". Each question used a 5-point Likert scale (ranging from "not at all" to "very much") to measure response. Higher scores indicated more work-life conflict.

## Data collection

Nurse manager' and nurse' surveys were conducted from August to September 2019. In the nurse manager survey, managers reported their unit type, the number of beds, and the average number of patients at midnight. These data were required to calculate the nurse staffing adequacy. Objective variables of the shiftwork characteristics of nurses were collected from the units' shiftwork schedules for the previous three months. This information was used to compute the average patients/beds-to-RN (registered nurse) ratio and other shiftwork characteristics. Nurses participated in the nurse survey by answering written questionnaires. We ensured that nurse managers were unable to identify whether individual nurses had participated or not. Data collection was undertaken after obtaining approval from the institutional review board (No. 1909/001-001). Documents containing information on the following were distributed: explanations of the purpose, methods, and procedure of the study; the confidentiality and anonymity of the data; and the fact participants could stop participating at any time for any reasons. Subsequently, participants' informed written consent to participate in the study was obtained. For analysis of shiftwork schedules per ward unit, to ensure anonymity, nurses' names were deleted from the unit shiftwork schedules by nurse' managers and replaced with serial numbers corresponding the shiftwork schedules and questionnaires.

## Data analysis

All collected data were analyzed using SPSS Statistics 22.0 (IBM Corp., Armonk, MY, USA). We described nurses' demographic characteristics, shiftwork characteristics, aspects of private life, and work-life conflict using real numbers, percentages, means, and standard deviations. The factors influencing work-life conflicts were analyzed using multiple linear regression analyses.

**Table 2. General characteristics of participants (n = 271).**

| Category | | Frequency | Percent |
|---|---|---|---|
| Sex | Male | 32 | 11.8 |
| | Female | 239 | 88.2 |
| Marital status | Single | 191 | 70.5 |
| | Married | 78 | 28.8 |
| | Divorced | 2 | 0.7 |
| Age (Mean 29.61) | 20–29 | 157 | 57.9 |
| | 30–39 | 88 | 32.5 |
| | ≥ 40 | 26 | 9.6 |
| Unit type | Medical ward | 80 | 29.5 |
| | Surgical ward | 20 | 7.4 |
| | Intensive care unit | 40 | 14.8 |
| | Emergency room | 48 | 17.7 |
| | Nursing care integrated services ward | 83 | 30.6 |
| Education | Associate degree | 94 | 34.8 |
| | Bachelor's degree | 172 | 63.7 |
| | Master's degree or higher | 4 | 1.5 |
| Work experience (Mean 6.64) | < 5 years | 137 | 50.6 |
| | 5≤~<10 years | 68 | 25.1 |
| | 10≤~<15 years | 35 | 12.9 |
| | 15≤~<20 years | 22 | 8.1 |
| | ≥ 20 years | 9 | 3.3 |
| Current work experience (Mean 4.76) | < 5 years | 174 | 64.2 |
| | 5≤~<10 years | 65 | 24.0 |
| | 10≤~<15 years | 18 | 6.6 |
| | 15≤~<20 years | 7 | 2.6 |
| | ≥ 20 years | 6 | 2.2 |
| Average income (Mean 2.47 Million Won) | Less than 2 Million Won | 18 | 6.6 |
| | Over 2~Less than 2.5 Million Won | 107 | 39.5 |
| | Over 2.5~Less than 3 Million Won | 99 | 36.5 |
| | Over 3 Million Won | 34 | 12.5 |
| | Non-response | 13 | 4.8 |

## Results

### General characteristics

Table 2 shows the general characteristics of 271 participating nurses. Most were female (88.2%) and more than 70% were single or unmarried. In terms of age, the largest proportion belonged to the 20–29 years group (57.9%). About 30% of the nurses worked in the nursing care integrated services ward; while 29.5%, 17.7%, 14.8%, and 7.4% worked in the medical, emergency, intensive care, surgical units, respectively. Approximately two-thirds held a baccalaureate or higher degree, and the participants had worked as an RN for 6.6 years on an average. In terms of monthly income, the majority (76%) earned between 2million and 3 million South Korean won.

### Shiftwork characteristics of nurses

The average unsocial work hours per week were 30.65 ± 5.33. The average number of unhealthy shift patterns was 4.36 ± 2.38. The overall nurse staffing adequacy was 100%, but

**Table 3. Results of shiftwork characteristics, aspects of private life, and work-life conflict.**

| Categories | Sub-categories | Variables | Mean ± SD/F (%) |
|---|---|---|---|
| Shiftwork characteristics | Work hours | Average number of over 40-hour weeks in a month | 2.02±0.98 |
| | | Unsocial working hours per week | 30.65±5.33 |
| | | Overtime work per day | 0.53±0.56 |
| | Shift pattern | Average number of maximum consecutive shifts per month | 6.59±2.01 |
| | | Average number of unhealthy shift patterns in a month | 4.36±2.38 |
| | Nurse staffing adequacy | Nurse staffing adequacy | 100% |
| | Shift control | Control over specific shifts | 2.38±0.69 |
| | | Control over specific start & finish times | 2.15±0.82 |
| | Shift instability | Required to change roster at short notice | 2.03±0.89 |
| | | Swapping shifts with colleagues | 1.89±0.75 |
| | | Requests to work specific shifts | 1.97±1.00 |
| Aspect of private life | Family status | Living alone | 59 (21.8) |
| | | Cohabiting with sponsor, friends, etc. | 39 (14.4) |
| | | Living with parents | 112 (41.3) |
| | | Living with children | 2 (0.7) |
| | | Living with sponsor, and children | 46 (17.0) |
| | | Living with parents, sponsor, and children | 13 (4.8) |
| | Demands at home | Dependent children | 61 (22.5) |
| | | Caring for someone with a chronic disease/handicap at home | 15 (5.5) |
| | | Caring for family/friend outside home | 22 (8.1) |
| | | Housekeeping responsibility | 120 (44.3) |
| | Life events during past year | Divorce | 2 (0.7) |
| | | Accidents | 7 (2.6) |
| | | Severe illness | 8 (3.0) |
| | | Death of important person | 16 (5.9) |
| | Time spent on housework/child rearing on workdays | | 1.64±1.81 |
| | Time spent on housework/child rearing on holidays | | 5.93±3.42 |
| | Support at home | Domestic help (available) | 179 (66.1) |
| | Leisure constraints | Intrapersonal leisure constraint | 2.73±0.64 |
| | | Interpersonal leisure constraint | 2.75±0.61 |
| | | Structural leisure constraint | 3.04±0.64 |
| Work-life conflict | Time based work-life conflict | Time based work interference with life | 3.23±0.85 |
| | | Time based life interference with work | 2.30±0.81 |

there were differences in each ward's degree of staffing. Nurses had low levels of control over shiftwork. The mean scores for the necessity of changes to the roster at short notice, swapping shifts with colleagues, and requests to work specific shifts were 2.03 ± 0.89, 1.89 ± 0.75, and 1.97 ± 1.00 (out of a total of 5), respectively (Table 3).

## Private lives of nurses

In terms of family status, a little less than half of the shift nurses were living with parents (41.3%). Regarding demands at home, close to half (44.3%) of all shift nurses had housekeeping responsibilities. The mean time spent on housework or childrearing on workdays and holidays was 1.64 ± 1.81 and 5.93 ± 3.42 hours, respectively. More than half of working nurses (66.1%) received domestic help. The means for intrapersonal, interpersonal, and structural leisure constraints were 2.73 ± 0.64, 2.75 ± 0.61, and 3.04 ± 0.64 (out of a total of 5), respectively (Table 3).

### Work-life conflict of nurses

The mean score for work-life conflict was 2.77 ± 0.68 (out of a total of 5); this means that participants of this study have a medium work-life conflict. The mean score for work interference with life was 3.23 ± 0.85, which was higher than the mean score of 2.30 ± 0.81 for life interference with work (Table 3).

### Factors influencing work-life conflict of nurses

Multiple regression analyses were conducted to identify the factors that independently related to work-life conflict. Table 4 shows the results of the regression analyses, which were conducted with a total of predictor variables: shiftwork characteristics and aspects of private life. Sex, marital status, and work experience were set as control variables. The regression model was found to be significant (F = 6.03, $p$ = < .001), and the adjusted coefficient of determination (Adj $R^2$), which indicates the explanatory power of the model, was .311. Significant factors related to work-life conflict included control over specific start and finish times of work (β = -0.16, $p$ = .019), frequency of swapping shifts with colleagues (β = -0.15, $p$ = .025), and leisure constraints (β = -0.39, $p$ = < .001).

## Discussions

This study reported that nurses worked over 40 hours a week in a month about twice a month as per their schedule. The nurses in our study reported unsocial working hours to cover an average of 30.65 hours per week. Unison [22] reported that unsocial working hours of nurses are divided into several categories, with Saturday nights and early morning hours paying an additional 50% of the basic pay, and Sundays or national holidays, earning nurses an additional 100% of the basic pay. However, regardless of the nature of unsocial working hours, only night and holiday working allowances are provided in Korea. The average amount of overtime per day was found to be 0.53 ± 0.56 hours. Lee, Jeong, Ko, and Kim [23] have revealed that the average amount of overtime was 1.32 hours. The occurrence of overtime can be seen as a problem associated with nurse staffing. Inappropriate nurse staffing will result in overtime work. The average number of unhealthy shift patterns was found to be 4.36 per month. It was frequently confirmed that nurses worked more than 3-shift rotations for more than five consecutive days, working with less than 24-hours of rest between shifts, working with less than two nurses on night shifts, a backward rotation working system, etc. A previous study found that a backward rotation working system results in a worse work-life balance [24]. In October 2019, guidelines issued for night shifts for nursing staff proposed a night shift planning, including guaranteeing rest (48 hours and over) between consecutive two-night shifts, and limiting periods of consecutive night shifts to less than three days [25]. However, current shiftwork schedules were found to be in contravention of these guidelines. Therefore, more detailed guidelines for shiftwork, particularly for night shifts, will have to be worked out.

Overall nurse staffing adequacy was found to be 100%. However, nurse staffing adequacy at the ward level showed significant variations from 33% to 162%. The differences among the wards were not only a matter of nurse staffing adequacy, but also contributed to other problems related to inappropriate nursing services, patient safety, the burden of workload, and turnover. Thus, it can be seen that the overall improvement in this area is necessary.

The scores for control over specific shifts and control over specific start and finish times were 2.38 and 2.15, respectively. Price [26] reported that 55% of respondents showed a low level of satisfaction when asked about the degree of control over the work environment. If welfare is strengthened with measures such as extra graded pay for shifts, providing at least 48

**Table 4. Relationship between shiftwork characteristics, aspect of private life, and work-life conflict.**

| Variables | work-life conflict | | | | |
|---|---|---|---|---|---|
| | B | SE | β | t | *p* |
| (Constant) | 1.61 | 0.41 | | 3.82 | < .001 |
| Sex | | | | | |
| Female (vs. male) | -0.14 | 0.12 | -0.07 | -1.17 | .242 |
| Marital status | | | | | |
| Married (vs. or not) | 0.43 | 0.18 | 0.29 | 2.46 | .015 |
| Work experience | -0.02 | 0.01 | -0.21 | -2.82 | .005 |
| **Shift work characteristics** | | | | | |
| Work hours | | | | | |
| Number of over 40-hour working weeks in a month | 0.04 | 0.05 | 0.06 | 0.89 | .376 |
| Unsocial working hours per week | -0.01 | 0.01 | -0.08 | -1.00 | .319 |
| Overtime | 0.13 | 0.07 | 0.11 | 1.93 | .054 |
| Shift pattern | | | | | |
| Number of maximum consecutive shifts over 3 months | -0.01 | 0.02 | -0.03 | -0.48 | .632 |
| Number of unhealthy shift patterns in a month | 0.03 | 0.02 | 0.09 | 1.04 | .299 |
| Nurse staffing adequacy | -0.24 | 0.15 | -0.09 | -1.61 | .110 |
| Shiftwork control | | | | | |
| Control over specific shifts | 0.09 | 0.07 | 0.09 | 1.32 | .188 |
| Control over specific start & finish times | -0.13 | 0.06 | -0.16 | -2.36 | .019 |
| Shift instability | | | | | |
| Required to change roster at short notice | 0.04 | 0.05 | 0.05 | 0.81 | .420 |
| Swapping shifts with colleagues | 0.14 | 0.06 | 0.15 | 2.26 | .025 |
| Requests to work specific shifts | 0.01 | 0.04 | 0.01 | 0.15 | .883 |
| **Aspect of private life** | | | | | |
| Family status (vs. living alone) | | | | | |
| Cohabiting with sponsor, friends, etc. | -0.13 | 0.16 | -0.07 | -0.83 | .408 |
| Living with parents | -0.12 | 0.11 | -0.09 | -1.12 | .265 |
| Living with children | -0.62 | 0.51 | -0.08 | -1.21 | .229 |
| Living with sponsor and children | 0.04 | 0.22 | 0.02 | 0.18 | .860 |
| Living with parents, sponsor and children | -0.19 | 0.25 | -0.06 | -0.78 | .438 |
| Demands at home | 0.02 | 0.05 | 0.03 | 0.31 | .756 |
| Time spent on housework/child rearing | 0.03 | 0.03 | 0.10 | 1.06 | .292 |
| Support at home (vs. yes) | 0.08 | 0.09 | 0.06 | 0.95 | .344 |
| Leisure constraints | 0.47 | 0.07 | 0.39 | 6.86 | < .001 |
| Adjusted $R^2$ = .311, F($p$) = 6.032 (< .001), VIF = 1.143~5.543 | | | | | |

Note: control variable (sex, marital status, work experience)

hours of rest between night shifts, and increasing the number of working nurses on night shifts, nurses will have the ability to choose among patterns of shifts that fit their situations. Before implementing the various shift types, the working environment should be restructured to give nurses more control. We found that the scores for the necessity for changes in the roster at short notice, that swapping of shifts with colleagues and requesting to work specific shifts were 2.03, 1.89, and 1.97, respectively. We were able to identify a high frequency of changes to the roster at short notice. Changing shifts with short notice could increase the burden of work on nurses, which could have negative effects on work-life balance. Unstable shifts mean that nurses do not possess enough resources to cope with unexpected work vacancies.

A prior study found that work interference with life was greater than life interference with work [27, 28]. Studies have also found that nurses who work longer hours are more likely to experience interference in life due to work [29]. That is, the more time spent on their work, the greater the experience of work interference in life. A previous study observed that working nights or on weekends interferes with family time, and it is one of the primary reasons for turnover [30]. Unsocial working hours, such as nights or weekend shifts, may interfere with family time, which may increase the degree of work interference with life. As nurses are overloaded with structural constraints, such as shiftwork, it is important to identify and improve factors that can reduce their work-life conflicts. A previous study reported that shiftwork can increase the chances of metabolic and cardiovascular disease and sleep disturbances [31]. Since, factors such as age and work experience cannot be altered, it is necessary to look for ways to reduce work-life conflict to improve the nurses' work environment and prevent turnover in clinical settings.

The lower degree of control over specific start and finish times of shifts makes for a higher degree of work-life conflict. Previous studies have also confirmed the findings of the current study in this regard [32]. It has been shown that when workers are allowed to control their working hours, the work environment and work-life balance improve [33]. While the direct comparison is difficult, this is similar to the fact that non-flexible working hours were shown to increase work-life conflicts [34] and that nurses who have flexible working hours find a better balance between work and life than those with non-flexible working hours [9]. Therefore, diverse shift patterns should be introduced into clinical nursing sites. The higher the frequency of shifts swapped with colleagues, the higher the degree of work-life conflict. Although no prior study was available for a comparison to regard to this finding, it can be surmised that the high frequency of shift changes with colleagues can be a burden to nurses, which is of a similar nature to the higher workload causing higher levels of work-life conflict [10].

This study reported that leisure constraints had an impact on work-life conflict. Previously, Park, Shin, and Shin [35] found that leisure balance was found to have a negative effect on work-life conflict, supporting the current findings. In particular, nurses in their late twenties with less than five years of work experience value leisure and work-life balance. However, in the actual working environment, shifts and weekend work greatly restrict leisure activities. The above-identified factors affecting work-life conflict should be appropriately incorporated into specific plans and policies for improving nursing environments in relation to shiftwork schedules. Nurses should be in control of the specific start and finish times of their shift schedules and thereby reduce the sudden change in their work schedules. Additionally, decreasing leisure constraints should be an important aspect aimed at improving the working conditions of the nursing staff management.

This study has several strengths. First, it has tried to provide a benefit to the management of nursing staff by identifying variables that can control the work-life conflict by studying shiftwork characteristics and aspects of private lives of nurses simultaneously. Second, to measure shiftwork characteristics, the ward schedules were analyzed directly to clarify which work patterns affect the work-life conflict. This attempt can serve as guidelines for further analysis of work schedules and finding ways to reduce work-life conflict. Third, leisure constraints were introduced as a variable related to the aspects of nurses' private lives and also proved to be a major variable to reduce the work-life conflict.

Nonetheless, this study has its limitations. First, the data obtained from cross-sectional survey and thus causality among the shiftwork characteristics, aspects of private life, and work-life conflict could not be determined. A longitudinal study evaluating the relationship among these factors could help extend findings in future studies. Second, this study selected four secondary hospitals using convenience extraction methods. It could be corrected by employing a larger sample size, or taking measures to ensure generalizability of the study.

## Conclusions

The implications of this study are as follows. First, prior studies that investigated the shiftwork characteristics of nurses mainly measured the subjective variables through nurses' perception. However, in this study, the number of 40-hour plus working weeks, unsocial working hours per week, the average number of unhealthy shift patterns, and nurse staffing adequacy were measured as objective variables by procuring the three months shiftwork schedules. It has minimized the recognition bias of nurses and complemented memory inaccuracies. Second, a total of eight unhealthy shift patterns were derived to measure the number of unhealthy shift patterns by analyzing the shiftwork schedules.

## Supporting information

**S1 File. Questionnaire (Korean).**
(DOCX)

**S2 File. Questionnaire (English).**
(DOCX)

## Author Contributions

**Conceptualization:** Hye-Kyung Oh, Sung-Hyun Cho.

**Data curation:** Hye-Kyung Oh.

**Formal analysis:** Hye-Kyung Oh.

**Investigation:** Hye-Kyung Oh.

**Methodology:** Hye-Kyung Oh.

**Writing – original draft:** Hye-Kyung Oh.

**Writing – review & editing:** Hye-Kyung Oh, Sung-Hyun Cho.

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
