## [Decision Letter · Decision Letter 0]

2 Sep 2020

PONE-D-20-18345

Effects of nurses’ shiftwork characteristics and aspects of private life on work-life conflict

PLOS ONE

Dear Dr. Oh,

Thank you for submitting your manuscript to PLOS ONE. After careful consideration, we feel that it has merit but does not fully meet PLOS ONE’s publication criteria as it currently stands. Therefore, we invite you to submit a revised version of the manuscript that addresses the points raised during the review process.

We look forward to receiving your revised manuscript.

Kind regards,

Ritesh G. Menezes, M.B.B.S., M.D., Diplomate N.B.

Academic Editor

PLOS ONE

Journal Requirements:

2. In your Methods section, please provide additional information about the participant recruitment method and the demographic details of your participants. Please ensure you have provided sufficient details to replicate the analyses such as: a) the recruitment date range (month and year), b) a description of any inclusion/exclusion criteria that were applied to participant recruitment, c) a table of relevant demographic details, d) a statement as to whether your sample can be considered representative of a larger population, e) a description of how participants were recruited, and f) descriptions of where participants were recruited and where the research took place.

4. Please provide additional details regarding participant consent. In the ethics statement in the Methods and online submission information, please ensure that you have specified (1) whether consent was informed and (2) what type you obtained (for instance, written or verbal). If your study included minors, state whether you obtained consent from parents or guardians. If the need for consent was waived by the ethics committee, please include this information.

Reviewers' comments:

Reviewer's Responses to Questions

**Comments to the Author**

1. Is the manuscript technically sound, and do the data support the conclusions?

Reviewer #1: Partly

Reviewer #2: Yes

Reviewer #3: Yes

Reviewer #4: Partly

2. Has the statistical analysis been performed appropriately and rigorously? 

Reviewer #1: Yes

Reviewer #2: Yes

Reviewer #3: Yes

Reviewer #4: Yes

3. Have the authors made all data underlying the findings in their manuscript fully available?

Reviewer #1: Yes

Reviewer #2: Yes

Reviewer #3: No

Reviewer #4: Yes

4. Is the manuscript presented in an intelligible fashion and written in standard English?

Reviewer #1: Yes

Reviewer #2: Yes

Reviewer #3: Yes

Reviewer #4: Yes

5. Review Comments to the Author

Reviewer #1: The authors investigate the effects of nurses' shiftwork schedules on their work-life balance.

It 's an interesting work that addresses a current and unresolved issue, i.e. the balancing of workers' health and well-being, hospitals' organization and inpatients' needs.

I have some suggestions and requests for minor revisions:

1) introduction:

- authors states that adjusting work schedules can be a solution to improve.... I suggest to enphazyze that in particular changes of the shift-patterns may make the difference (cite Wilson JL. Nurs. Manag. 2002, 10. 211-219; Barton J et al. Occup. Envirom. Med. 1994, 51, 749-755, and Shiffer D et al Int. J Environ Res Public Health. 2018 Sep 18;15(9):2038. doi: 10.3390/ijerph15092038h) these same citations should be mentioned throughout the discussion when appropriate

- typos: children and time spent (not spend) on domestic...

- state more clearly the aim of the study at the end of introduction

2) variables and ...

- you mention figure 1 that I can't see

3) discussion

- the first sentence is not clear, please rephrase

- in England, ....it is an examples

- add some thoughts about the increased cv risk due to shifts, work-life conflicts, the impact of such conflicts on worse patients' care

- try to put the results in perspectives

Reviewer #2: Thank you for the opportunity to review this article, titled “Effects of nurses’ shiftwork characteristics and aspects of private life on work-life conflict.” This is an overall well-written paper. However, I think it could be a better article with a few modifications, so I'll give you some opinions.

My suggestions for revision are:

1. Results

Please check the number in the first sentence. 217? 271? Which one is correct?

2. Discussion

What does 2.02 mean in the first sentence? Please clarify the meaning.

3. Conclusion

It is good to articulate your findings, but it doesn't seem necessary to repeat them in your conclusions. Your implications are seen as the strength of this paper. At the end of the discussion, describe the strengths and limitations of this paper. In the conclusion, the implications will be described, but based on the results of this study, please describe in more detail the direction of plans and policies for improving nursing environments that the authors want to propose.

Thank you for your writing a good paper.

Reviewer #3: 1. The manuscript is technically sound and well prepared. The results support the conclusion

2. Statistical analysis is performed appropriately as per the objectives of the study

3. Data collection scales are made available whereas all data underlying the findings in the manuscript is not fully

available.

4. Manuscript is well written, however there are few errors noted which needs to be rectified. Comments are given in

the manuscript.

5.

• Setting and sample: Author has mentioned that the data collection was done from 4 hospitals.

There could be mention about the type of the hospital (tertiary care hospitals).

• Sampling techniques adopted for the study is not mentioned. As the study was multicentric what proportion of nurses

were selected from each of the hospital need to be mentioned. Did the samples represent from all four hospitals

needs to be mentioned.

• The sample size calculation needs to be mentioned.

• Data collection: The study includes shift nurses working in the specified areas of the four hospitals as mentioned under the setting and sample section, whereas under data collection it is mentioned that 'Nurse manager’ and nurse’ surveys were conducted'. There is no data presented on the nurse managers. The reason for conducting survey among nurse managers to be specified.

• There is mention on approval for IRB but process of taking informed consent from the participants needs to be included.

• As the study aims to find the shift work schedule, the reader has no information on the shift timing across the hospitals in the country. This could be included.

• Variables and instruments: All the instruments need to be described including the scoring and interpretation. Eg work-life conflict scale needs mention on number of items in the scale and interpretation.

• Results: Results are presented as per the objectives. There were few errors noted needs to be rectified.

• Discussion: Compared the results with the previous study. Directs to draw the conclusions.

• Conclusion: Limitations of the study needs to be mentioned.

Reviewer #4: Indeed, this study is a worth study in measuring the nurses’ shift work characteristics and aspects of private life on work-life conflict.

I would like to suggest the following recommendations to improve the scientific rigor of the study.

1. Mention the appropriate sample size calculation formula used to calculate sample size.

2. In the study it is mentioned that samples were selected based on their willingness other than that have you used any inclusion and exclusion criteria for sample selection.

3. Mention the sampling method.

4. The authors made a good attempt to describe the variables, instruments and their validity.

5. The discussion part appears to be confused, please try to organise the discussion part same as that of the results, by including more studies.

6. Mention the strength and weakness of your study.

7. Try to include the limitations.

8. Write the conclusion properly.

9. Do a language editing.

6. PLOS authors have the option to publish the peer review history of their article (what does this mean?). If published, this will include your full peer review and any attached files.

Reviewer #1: **Yes: **Laura Adelaide Dalla Vecchia

Reviewer #2: No

Reviewer #3: No

Reviewer #4: **Yes: **Dr Shejila Chillakunnel Hussain Rawther

---

## [Author Response · Author response to Decision Letter 0]

30 Sep 2020

Reviewer #1: 

1. Introduction

-author states that adjusting work schedules can be a solution to improve. I suggest to emphasize that in particular changes of the shift-patterns may make the difference (cite Wilson JL. Nurs. Manag. 2002, 10. 211-219; Barton J et al. Occup. Envirom. Med. 1994, 51, 749-755, and Shiffer D et al Int. J Environ Res Public Health. 2018 Sep 18;15(9):2038. doi: 10.3390/ijerph15092038h) these same citations should be mentioned throughout the discussion when appropriate

- typos: children and time spent (not spend) on domestic...

- state more clearly the aim of the study at the end of introduction

Author’s Response: 

- We really appreciate the citation lists you have provided us with. We have added the Shiffer et al. (2018) citation in the Discussion section.

- We have corrected the typos (page 4. Line 74).

- We have added the aim of the study at the end of introduction:

‘the general aim was to identify the factors affecting work-life conflict of nurses. Taking these factors into account, we recommend that plans and policies should be implemented to improve the environment of the three-shift schedules of the nurses.’ (page 4. Lines 77-80).

2. variables and ...

- you mention figure 1 that I can't see

Author’s Response: 

- We apologize for the inconvenience caused We have attached the individual Figure 1 image file.

3) discussion

- the first sentence is not clear, please rephrase

Author’s Response: 

- We have changed the first sentence:

‘This study reported that nurses worked over 40-hours a week about twice a month as per their schedules’ (page 17. Lines 248-249).

- in England, ....it is an examples

Author’s Response: 

- We have changed the order of citation.

‘Unison [21] reported that unsocial working hours of nurses are divided into several categories, with Saturday nights and early morning hours paying an additional 50% of the basic pay, and Sundays or national holidays, earning nurses an additional 100% of the basic pay.’ (page 17. Line 250-253).

- add some thoughts about the increased cv risk due to shifts, work-life conflicts, the impact of such conflicts on worse patients' care

Author’s Response: 

- We have added relevant thoughts on the negative aspects of shiftwork. 

‘A previous study reported that shiftwork can increase the chances of metabolic and cardiovascular disease and sleep disturbances.’ (Page 19, Lines 296-298).

- try to put the results in perspectives

Author’s Response: 

- We have presented in results as per the study objectives.

 

Reviewer #2: 

1. Results

- Please check the number in the first sentence. 217? 271? Which one is correct?

Author’s Response: 

- 271 is the correct number. We have corrected the number of participants (Page 10, Line 193).

2. Discussion

- What does 2.02 mean in the first sentence? Please clarify the meaning.

Author’s Response: 

- We have changed the first sentence:

‘This study reported that nurses worked over 40-hour a week about twice a month, as per their schedule’ (page 17. Lines 248-249).

3. Conclusion

- It is good to articulate your findings, but it doesn't seem necessary to repeat them in your conclusions. Your implications are seen as the strength of this paper. At the end of the discussion, describe the strengths and limitations of this paper. In the conclusion, the implications will be described, but based on the results of this study, please describe in more detail the direction of plans and policies for improving nursing environments that the authors want to propose.

Author’s Response: 

- We have removed the repetitive sections in conclusions and added the strengths and limitations of this study. (Page 20, Lines 325-338). We have described in details the direction of plans and policies. The following lines have been added.

‘Nurses should be in control of the specific start and finish times of their shift schedules and thereby reduce the sudden change in their work schedules. Additionally, decrease leisure constraints should be an important aspect aimed at improving the working conditions of the nursing staff management.’ (Pages 20, Lines 321-324).

Reviewer #3: 

1. The manuscript is technically sound and well prepared. The results support the conclusion

2. Statistical analysis is performed appropriately as per the objectives of the study

Author’s Response: Thank you for your comments!

3. Data collection scales are made available whereas all data underlying the findings in the manuscript is not fully

available.

Author’s Response:

- We have added all the relevant data underlying the finding. 

4. Manuscript is well written, however there are few errors noted which needs to be rectified. Comments are given in the manuscript.

Author’s Response: 

- We have adjusted the manuscript according your comments. Thank you for your detailed comments! According your recommendations, the following sections of the manuscript have been adjusted (Page 5, Lines 102-103, Pages 6-7, Lines 130-132, Page 9, Line 152, Page 9, Lines160-163, Page 9, Lines 168-169, Page 13, Lines 215, 219).

- This study reported an average of 2.02 over 40-hour weeks in a month.

Prior to this sentence, you mentioned that this should be written as mean work hours. After completing this study, our researchers also arrived at same conclusion, in alignment with your comments, therefore, we will employ a counting method base on your recommendations in another research.

- However, nurse staffing adequacy at the ward level showed significant variations from 33% to 162%.

Prior to this sentence, you mentioned that this value needs to be rechecked as it exceeds more than 100. There may be more than 100 % of nursing staff adequacy per unit. Because we had checked the current recruitment rate of nurses compared to nurses needed per unit.

5-1.

• Setting and sample: Author has mentioned that the data collection was done from 4 hospitals.

There could be mention about the type of the hospital (tertiary care hospitals).

Author’s Response: 

- We have specified the type of the hospital:

‘Convenience samples of 271 shift nurses were selected from four secondary care hospitals with medical, surgical, intensive care, emergency, and integrated nursing care units (Page 5, Lines 85-86).

5-2. • Sampling techniques adopted for the study is not mentioned. As the study was multicentric what proportion of nurses were selected from each of the hospital need to be mentioned. Did the samples represent from all four hospitals needs to be mentioned.

Author’s Response: 

- We have added the sampling techniques and mentioned the limitations of representation:

‘Convenience samples of 271 shift nurses selected from 4 secondary care hospitals with medical, surgical, intensive care, emergency, and integrated nursing care units. The number of selected ward types was chosen according to the hospital’s circumstance. The participants were recruited per ward unit with the cooperation of the nursing departments and hospitals’ (Page 5, Lines 85-89). 

‘Second, this study selected four secondary hospitals using convenience extraction methods.’ (Pages 20, Lines 336-337). 

5-3 • The sample size calculation needs to be mentioned.

Author’s Response: 

We have added the sample size calculation method.

‘The sample size was calculated using G*power 3.0.10 based on a significance level of 0.05, a medium effect size (0.15), a power of 0.95, and 11 independent variables. According to this analysis, the minimum sample size was 178, although 280 were ultimately recruited with consideration of the dropout rate.’ (Page 5, Lines 93-96).

5-4 • Data collection: The study includes shift nurses working in the specified areas of the four hospitals as mentioned under the setting and sample section, whereas under data collection it is mentioned that 'Nurse manager’ and nurse’ surveys were conducted'. There is no data presented on the nurse managers. The reason for conducting survey among nurse managers to be specified.

Author’s Response: 

- We have added the following sentence:

‘These data were required to calculate the nurse staffing adequacy.’ (Page 10, Lines 168-169).

5-5 • There is mention on approval for IRB but process of taking informed consent from the participants needs to be included.

Author’s Response: 

- We have added relevant details the process of procuring informed consent from the participants.

‘Documents containing information on the following were distributed: explanations of the purpose, methods, and procedure of the study; the confidentiality and anonymity of the data; and the fact that the participants could stop participating at any time, for any reasons. Subsequently, participants’ informed written consent to participate in the study was obtained.’ (Page 10, Lines 175-179).

5-6 • As the study aims to find the shift work schedule, the reader has no information on the shift timing across the hospitals in the country. This could be included.

Author’s Response: 

- We have added the following sentence:

‘In Korea, hospital nurses work three shifts [1]. The shifts are classified into days, afternoons, and nights.’ (Page 3, Lines 40-41).

5-7 • Variables and instruments: All the instruments need to be described including the scoring and interpretation. Eg work-life conflict scale needs mention on number of items in the scale and interpretation.

Author’s Response: 

- We have added information on the scoring and interpretation of the work-life conflict (Page 10, Lines 160-163).

• Results: Results are presented as per the objectives. There were few errors noted needs to be rectified.

Author’s Response: 

- We have adjusted the results according your comments. Thank you for your detailed comments!

• Discussion: Compared the results with the previous study. Directs to draw the conclusions.

Author’s Response: 

- We have added the relevant studies in the discussion. 

• Conclusion: Limitations of the study needs to be mentioned.

Author’s Response: 

- We have added the limitation of this study:

‘Nonetheless, this study has its limitations. The data were obtained from a cross-sectional survey and thus causality among the shiftwork characteristics, aspects of private life, and work-life conflict could not be determined. A longitudinal study evaluating the relationship among these factors could help extend findings in future studies. Second, this study selected four secondary hospitals using convenience extraction methods. It could be corrected by employing a larger sample size, or taking measures to ensure generalizability of the study’ (Page 20, Lines 333-338). 

Reviewer #4: Major comments

1. Mention the appropriate sample size calculation formula used to calculate sample size.

Author’s Response: 

- We have added the sample size calculation method.

‘The sample size was calculated using G*power 3.0.10 based on a significance level of 0.05, a medium effect size (0.15), a power of 0.95, and 11 independent variables. According to this analysis, the minimum sample size was 178, although 280 were ultimately recruited with consideration for the dropout rate.’ (Pages 5, Lines 93-96).

2. In the study it is mentioned that samples were selected based on their willingness other than that have you used any inclusion and exclusion criteria for sample selection.

Author’s Response: 

- We have added the inclusion and exclusion criteria as follows:

‘The inclusionary criteria involved (a) shiftwork nurses, (b) who working at the designated hospitals. Nurses were excluded if they were newly recruited, under training, and could not work independently’ (Page 5, Lines 90-93).

3. Mention the sampling method.

Author’s Response: 

- We have added the sampling method:

‘Convenience samples of 271 shift nurses selected from four secondary care hospitals’ (Page 5, Line 85).

4. The authors made a good attempt to describe the variables, instruments and their validity.

Author’s Response: 

- Thank you for your comment!

5. The discussion part appears to be confused, please try to organize the discussion part same as that of the results, by including more studies.

Author’s Response: 

- We have included more studies in the discussion. 

6. Mention the strength and weakness of your study.

Author’s Response: 

- We have added the strengths and limitations of this study in the Discussion section (Page 20, Lines 325-338).

7. Try to include the limitations.

Author’s Response: 

- We have added the limitation of this study as follows:

‘Nonetheless, this study has its limitations. The data were obtained from a cross-sectional survey and thus causality among the shiftwork characteristics, aspects of private life, and work-life conflict could not be determined. A longitudinal study evaluating the relationship among these factors could help extend findings in future studies. Second, this study selected four secondary hospitals using convenience extraction methods. It could be corrected by employing a larger sample size, or taking measures to ensure generalizability of the study’ (Page 20, Lines 333-338).

8. Write the conclusion properly.

Author’s Response: 

- We have removed the repetitive sections of the conclusions and added the strengths and limitations of this study. (Page 20, Lines 325-338).

9. Do a language editing.

Author’s Response: 

- We agree with the reviewer’s advice and have made substantial revisions accordingly. We also had the manuscript rechecked by a native English speaker. Consequently, many grammatical and stylistic edits have been made throughout the manuscript. We hope the revised manuscript meets your expectations.

---

## [Decision Letter · Decision Letter 1]

2 Nov 2020

Effects of nurses’ shiftwork characteristics and aspects of private life on work-life conflict

PONE-D-20-18345R1

Dear Dr. Oh,

We’re pleased to inform you that your manuscript has been judged scientifically suitable for publication and will be formally accepted for publication once it meets all outstanding technical requirements.

Kind regards,

Ritesh G. Menezes, M.B.B.S., M.D., Diplomate N.B.

Academic Editor

PLOS ONE

Reviewers' comments:

Reviewer's Responses to Questions

**Comments to the Author**

1. If the authors have adequately addressed your comments raised in a previous round of review and you feel that this manuscript is now acceptable for publication, you may indicate that here to bypass the “Comments to the Author” section, enter your conflict of interest statement in the “Confidential to Editor” section, and submit your "Accept" recommendation.

Reviewer #1: All comments have been addressed

Reviewer #3: All comments have been addressed

Reviewer #4: All comments have been addressed

2. Is the manuscript technically sound, and do the data support the conclusions?

Reviewer #1: Yes

Reviewer #3: Yes

Reviewer #4: Yes

3. Has the statistical analysis been performed appropriately and rigorously? 

Reviewer #1: Yes

Reviewer #3: Yes

Reviewer #4: Yes

4. Have the authors made all data underlying the findings in their manuscript fully available?

Reviewer #1: Yes

Reviewer #3: Yes

Reviewer #4: Yes

5. Is the manuscript presented in an intelligible fashion and written in standard English?

Reviewer #1: Yes

Reviewer #3: Yes

Reviewer #4: Yes

6. Review Comments to the Author

Reviewer #1: (No Response)

Reviewer #3: Author has incorporated the suggestions given and has thoroughly revised the manuscript. Details on sampling technique, obtaining consent and data collection method is given. Limitations of the study are included.

Reviewer #4: (No Response)

7. PLOS authors have the option to publish the peer review history of their article (what does this mean?). If published, this will include your full peer review and any attached files.

Reviewer #1: **Yes: **Laura Adelaide Dalla Vecchia

Reviewer #3: **Yes: **Prima Jenevive Jyothi D'Souza

Reviewer #4: **Yes: **Shejila C Rawther

---

## [Editor Report · Acceptance letter]

18 Nov 2020

PONE-D-20-18345R1 

Effects of nurses’ shiftwork characteristics and aspects of private life on work-life conflict 

Dear Dr. Oh:

I'm pleased to inform you that your manuscript has been deemed suitable for publication in PLOS ONE. Congratulations! Your manuscript is now with our production department. 

Kind regards, 

on behalf of

Prof. Dr. Ritesh G. Menezes 

Academic Editor

PLOS ONE